# Solvent Fractionation of *Polygonum cuspidatum* Sieb. et Zucc. for Antioxidant, Biological Activity, and Chromatographic Characterization

**DOI:** 10.3390/ijms26147011

**Published:** 2025-07-21

**Authors:** Yuchen Cheng, Yuri Kang, Woonjung Kim

**Affiliations:** Department of Chemistry, Hannam University, Daejeon 305-811, Republic of Korea; 000cheng@naver.com (Y.C.); yuri_1005@nate.com (Y.K.)

**Keywords:** *Polygonum cuspidatum* Sieb. et Zucc., antioxidant activity, enzyme inhibitory activity, chromatographic analyses

## Abstract

This study investigated the natural bioactive compounds in *Polygonum cuspidatum* Sieb. et Zucc. (*P. cuspidatum*) by fractionating a 70% ethanol extract using *n*-hexane, chloroform, ethyl acetate, *n*-butanol, and water. The total polyphenol and flavonoid contents of each fraction were determined, and their antioxidant activities were evaluated using DPPH, ABTS, and FRAP assays. Additionally, the anti-diabetic potential was assessed via α-glucosidase inhibitory activity, while anti-obesity activity was evaluated using lipase inhibitory activity. The fractions were also tested for tyrosinase and elastase inhibitory activities to assess their skin-whitening and anti-wrinkle potential, and their antibacterial activity against *Staphylococcus aureus*, *Escherichia coli*, and *Pseudomonas aeruginosa* was determined using the agar diffusion method. Finally, bioactive compounds were identified and quantified using HPLC and GC–MSD. The results showed that the ethyl acetate fraction possessed the highest total polyphenol content (0.53 ± 0.01 g GAE/g) and total flavonoid content (0.19 ± 0.02 g QE/g). It also exhibited strong antioxidant activity, with the lowest DPPH radical scavenging IC_50_ (0.01 ± 0.00 mg/mL), ABTS radical scavenging IC_50_ (0.06 ± 0.00 mg/mL), and the highest FRAP value (6.02 ± 0.30 mM Fe^2+^/mg). Moreover, it demonstrated potent enzyme inhibitory activities, including tyrosinase inhibitory activity (67.78 ± 2.50%), elastase inhibitory activity (83.84 ± 1.64%), α-glucosidase inhibitory activity (65.14 ± 10.29%), and lipase inhibitory activity (85.79 ± 1.04%). In the antibacterial activity, the ethyl acetate fraction produced a clear inhibitory zone of 19.50 mm against *Staphylococcus aureus*, indicating notable antibacterial activity. HPLC-PDA and GC–MSD analyses identified tannic acid and emodin as the major bioactive constituents. These findings suggest that the ethyl acetate fraction of *P. cuspidatum* extract, rich in polyphenol and flavonoid compounds, is a promising natural source of bioactive ingredients for applications in the food, pharmaceutical, and cosmetic industries. Further research is needed to explore its mechanisms and therapeutic applications.

## 1. Introduction

*Polygonum cuspidatum* Sieb. et Zucc. (*P. cuspidatum*), a perennial herb of the *Polygonaceae* family, is widely distributed across East Asia, including China, Japan, and Korea [1]. The dried rhizomes of this plant, known as *P. cuspidatum*, have been traditionally used in Chinese medicine for their effects in clearing heat and detoxifying, promoting diuresis, removing jaundice, and activating blood circulation to remove blood stasis. Clinically, it is commonly employed to treat liver and gallbladder diseases, rheumatic pain, and menstrual disorders [2]. With increasing research into natural medicines and plant resources, the active ingredients of *P. cuspidatum* and their multitarget pharmacological mechanisms have become a research focus. Current studies have shown that *P. cuspidatum* is rich in anthraquinones, such as emodin, chrysophanol, and rhein; stilbenes like resveratrol; flavonoids such as quercetin, kaempferol, and isoquercitrin; terpenoids like β-sitosterol and campesterol; as well as tannins and polyphenols [3,4,5,6]. These components have demonstrated significant antioxidant, anti-inflammatory, antibacterial, anticancer, metabolic-regulating, and skin-whitening activities in both in vitro and in vivo experiments [7,8]. Among these, antioxidant activity is considered the core mechanism underlying the diverse pharmacological effects of *P. cuspidatum*. Its primary mechanisms include the clearance of excessive reactive oxygen species (ROS), enhancement of superoxide dismutase (SOD) and glutathione (GSH) activities, and inhibition of lipid peroxidation, which significantly alleviates oxidative damage in HepG2 cells [9]. Moreover, resveratrol, an active compound in *P. cuspidatum*, induces autophagy and inhibits α-MSH induced melanogenesis, thereby achieving a whitening effect [10]. Furthermore, *P. cuspidatum* shows significant potential in natural medicine research. Its major active compounds, including resveratrol, emodin, and polydatin, inhibit xanthine oxidase activity, reduce blood uric acid levels, and exhibit antihyperuricemic effects while regulating the onset and pathological progression of chronic metabolic diseases through multiple mechanisms [11]. The highly diverse chemical composition of *P. cuspidatum* leads to the presence of numerous inactive impurities in its crude extracts, potentially affecting the accuracy of biological activity studies. Over the past few decades, research on the extraction of phenolic compounds from plants has primarily focused on advanced purification techniques such as solid-phase extraction (SPE), countercurrent chromatography (CCC), and centrifugal partition chromatography (CPC). However, these methods, while efficient in compound separation, are often complicated and associated with high operational costs [12,13]. In contrast, solvent fractionation has been widely applied in natural product research due to its efficiency, ease of operation, and low cost, particularly in polarity classification and preliminary purification [14]. Therefore, this study employed solvent fractionation, sequentially using *n*-hexane, chloroform, ethyl acetate, *n*-butanol, and water, to achieve the preliminary separation of components with different polarities. This method reduces interference from inactive impurities and improves the purity of the target compounds, providing a more reliable foundation for subsequent biological activity evaluation and component analysis.

## 2. Results

### 2.1. Total Polyphenol and Flavonoid Contents of Solvent Fractions from P. cuspidatum

Phenolic compounds are important secondary metabolites widely present in plants, commonly found in fruits, vegetables, grains, and medicinal herbs. The phenolic hydroxyl groups (–OH) in their structures enable them to scavenge reactive oxygen species (ROS), thereby mitigating oxidative damage, delaying aging, and promoting overall health [15]. This class includes flavonoids, anthocyanins, tannins, catechins, isoflavones, lignans, and resveratrol, all of which exhibit diverse biological activities such as antioxidant, antibacterial, anticancer, hepatoprotective, and lipid-lowering effects. Among them, flavonoids, characterized by a flavone backbone, are key pigments in leaves, flowers, fruits, and stems and play multiple roles in plants by absorbing ultraviolet radiation, defending against pathogens and herbivores, regulating growth and development, and modulating enzyme activity [16]. In this study, we measured the total polyphenol content in the solvent fractions of *P. cuspidatum* using a gallic acid standard curve, and the results are presented in Table 1. The ethyl acetate fraction exhibited the highest total polyphenol content (0.53 ± 0.01 g GAE/g), followed by the chloroform fraction (0.37 ± 0.01 g GAE/g), the *n*-butanol fraction (0.36 ± 0.02 g GAE/g), the *n*-hexane fraction (0.12 ± 0.00 g GAE/g), and the aqueous fraction (0.10 ± 0.01 g GAE/g), showing a statistically significant decreasing trend (*p* < 0.05). Additionally, we measured the total flavonoid content in the solvent fractions using a quercetin standard curve, and these results are also presented in Table 1. The ethyl acetate fraction exhibited the highest total flavonoid content (0.19 ± 0.02 g QE/g), followed by the *n*-butanol fraction (0.14 ± 0.01 g QE/g), the chloroform fraction (0.09 ± 0.00 g QE/g), the *n*-hexane fraction (0.08 ± 0.00 g QE/g), and the aqueous fraction (0.04 ± 0.00 g QE/g), also showing a statistically significant decreasing trend (*p* < 0.05). The significantly higher total polyphenol and flavonoid contents in the ethyl acetate fraction can be primarily attributed to its intermediate polarity, which offers optimal solubility and strong affinity for moderately polar phenolic compounds containing both hydrophilic hydroxyl groups and hydrophobic aromatic rings [17]. These findings indicate that ethyl acetate is the most effective solvent for extracting polyphenolic compounds, including flavonoids, from *P. cuspidatum*, making it particularly suitable for the isolating of bioactive phenolic constituents. Moreover, previous studies on medicinal plants such as *Curcuma longa* (turmeric) have also demonstrated the high efficiency of ethyl acetate in extracting phenolic and flavonoid compounds, further supporting the present findings [18]. This reinforces the role of ethyl acetate as a medium-polarity solvent capable of selectively enriching valuable phytochemicals and underscores the importance of solvent selection based on compound polarity in the development of natural products and strategies for functional ingredient extraction.

### 2.2. Antioxidant Activities (DPPH, ABTS Radical Scavenging Activity, and Ferric Reducing Antioxidant Power (FRAP) Assay) of Solvent Fractions from P. cuspidatum

The DPPH radical scavenging assay is a widely used in vitro method to evaluate the antioxidant capacity of compounds as it reflects the ability of antioxidants to donate hydrogen atoms or electrons to neutralize free radicals [19]. This method is based on the ability of antioxidants to scavenge stable DPPH radicals. The ABTS radical scavenging assay assesses antioxidant capacity by measuring the neutralization of ABTS^+^ radicals generated by potassium persulfate [20]. The FRAP assay quantifies the total reducing power of a sample by evaluating the ability of antioxidants to reduce ferric-TPTZ (Fe^3+^-TPTZ) to ferrous-TPTZ (Fe^2+^-TPTZ) [21]. The DPPH radical scavenging activity of solvent fractions from *P. cuspidatum*, expressed by the IC_50_ values, is presented in Table 2. The ethyl acetate fraction and L-ascorbic acid exhibited the highest antioxidant activities (0.01 ± 0.00 mg/mL), followed by the *n*-butanol fraction (0.02 ± 0.00 mg/mL), the chloroform fraction (0.07 ± 0.00 mg/mL), the aqueous fraction (0.10 ± 0.01 mg/mL), and the *n*-hexane fraction (0.25 ± 0.20 mg/mL), showing a statistically significant decreasing trend (*p* < 0.05). The ABTS radical scavenging activity of solvent fractions from *P. cuspidatum*, expressed by the IC_50_ values, is presented in Table 2. The ethyl acetate fraction and L-ascorbic acid exhibited the highest antioxidant activities (0.06 ± 0.00 mg/mL), followed by the *n*-butanol fraction (0.13 ± 0.00 mg/mL), the chloroform fraction (0.14 ± 0.00 mg/mL), the aqueous fraction (0.51 ± 0.00 mg/mL), and the *n*-hexane fraction (0.64 ± 0.19 mg/mL), showing a statistically significant decreasing trend (*p* < 0.05). The ferric reducing antioxidant power (FRAP) assay of solvent fractions from *P. cuspidatum* is presented in Table 2. The ethyl acetate fraction exhibited the highest antioxidant activity (6.02 ± 0.30 mM Fe^2+^/mg), followed by the *n*-butanol fraction (2.95 ± 0.57 mM Fe^2+^/mg), the chloroform fraction (1.84 ± 0.14 mM Fe^2+^/mg), the aqueous fraction (0.32 ± 0.02 mM Fe^2+^/mg), and the *n*-hexane fraction (0.22 ± 0.05 mM Fe^2+^/mg), showing a statistically significant decreasing trend (*p* < 0.05). The differences in antioxidant activity among the solvent fractions of *P. cuspidatum* can primarily be attributed to variations in the electron or hydrogen donation capabilities of phenolic compounds, which play a crucial role in scavenging free radicals and reducing oxidants [22]. Polyphenolic compounds are well known for their redox properties, effectively neutralizing reactive species by donating electrons or hydrogen atoms, thereby interrupting oxidative chain reactions and protecting biological macromolecules from oxidative damage [23]. Fractions extracted using medium-polarity solvents, such as ethyl acetate and *n*-butanol, exhibited higher antioxidant activities, indicating that these solvents can selectively enrich compounds with strong redox activity. Moreover, similar solvent polarity-dependent extraction profiles have been reported in other medicinal plants, such as *Leucaena leucocephala*, where ethyl acetate demonstrated high efficiency in extracting polyphenolic and flavonoid compounds [24]. These bioactive compounds typically contain hydroxyl groups attached to aromatic rings, which confer superior free radical scavenging abilities. Collectively, these findings underscore the importance of rational solvent selection based on polarity to enhance the extraction efficiency and quality of natural antioxidants from plant materials.

### 2.3. Tyrosinase and Elastase Inhibitory Activities of Solvent Fractions from P. cuspidatum

Tyrosinase plays a crucial role in regulating skin pigmentation by mediating melanin synthesis and is a key target for improving skin health [25]. This study evaluated the inhibitory effects of various solvent fractions from *P. cuspidatum* on tyrosinase activity. As shown in Table 3, at a concentration of 0.1 mg/mL, the positive control, Kojic acid, exhibited an inhibitory activity of 69.26 ± 0.10%. Among the solvent fractions, the ethyl acetate fraction exhibited the highest inhibitory activity, with a rate of 67.78 ± 2.50%, followed by the *n*-butanol fraction (61.64 ± 0.65%), chloroform fraction (59.93 ± 1.95%), aqueous fraction (56.31 ± 0.86%), and *n*-hexane fraction (53.60 ± 1.61%), showing a statistically significant decreasing trend (*p* < 0.05). Ethyl acetate, as a medium-polar solvent, effectively extracts medium-polar compounds from *P. cuspidatum*, such as resveratrol, emodin, and piceid. These compounds are closely associated to tyrosinase inhibitory activity [26]. The polarity of ethyl acetate allows it to interact effectively with the structures of these compounds, thereby enhancing their bioactivity and resulting in the strongest enzyme inhibitory activity [27]. Consequently, the ethyl acetate fraction demonstrates significant potential for skin whitening, particularly in applications aimed at inhibiting melanin production.

Elastase is an enzyme responsible for the degradation of elastin, which plays a critical role in maintaining skin elasticity. This study evaluated the inhibitory effects of different solvent fractions from *P. cuspidatum* on elastase activity. As shown in Table 3, at a concentration of 0.1 mg/mL, the positive control, L-ascorbic acid, exhibited an inhibitory activity of 63.64 ± 0.59%. The ethyl acetate fraction exhibited the strongest elastase inhibitory activity, reaching 83.84 ± 1.64%, followed by the *n*-butanol fraction (74.75 ± 1.74%), aqueous fraction (72.2 ± 0.60%), *n*-hexane fraction (68.69 ± 1.31%), and chloroform fraction (64.65 ± 1.50%), with a statistically significant decreasing trend (*p* < 0.05). The potent elastase inhibitory activity of the ethyl acetate fraction may be attributed to its content of medium-polar compounds, such as resveratrol, emodin, and piceid. High-performance liquid chromatography coupled with mass spectrometry (HPLC-Q-TOF-MS/MS) [28] has shown that these compounds are closely associated with elastase inhibitory activity. Therefore, the ethyl acetate fraction not only holds potential for skin whitening but also plays a significant role in anti-aging and maintaining skin elasticity.

### 2.4. α-Glucosidase and Lipase Inhibitory Activities of Solvent Fractions from P. cuspidatum

α-Glucosidase and lipase are involved in the digestion and absorption of carbohydrates and lipids, respectively, and their enzymatic activities are closely associated with metabolic disorders such as obesity and type 2 diabetes. The inhibitory activity of α-glucosidase can delay glucose absorption and reduce postprandial blood glucose levels, while lipase inhibitory activity helps decrease dietary fat absorption. Both mechanisms are considered effective strategies for regulating metabolic health [29,30]. The α-glucosidase inhibitory activity of solvent fractions from *P. cuspidatum* is presented in Table 4. At the test concentration of 0.1 mg/mL, the positive control acarbose showed 34.83 ± 0.31% inhibitory activity. Among the fractions, the ethyl acetate fraction exhibited the highest inhibitory activity (65.14 ± 0.29%), followed by the *n*-butanol fraction (35.87 ± 0.56%), the chloroform fraction (12.87 ± 0.15%), the aqueous fraction (6.46 ± 0.40%), and the *n*-hexane fraction (5.56 ± 0.59%), showing a statistically significant decreasing trend (*p* < 0.05). The lipase inhibitory activity of solvent fractions from *P. cuspidatum* is presented in Table 4. At the test concentration of 0.1 mg/mL, the positive control orlistat showed 98.18 ± 0.18% inhibitory activity. Among the fractions, the *n*-hexane fraction exhibited the highest inhibitory activity (85.93 ± 1.00%), followed by the ethyl acetate fraction (85.79 ± 0.61%), the *n*-butanol fraction (84.35 ± 0.77%), the aqueous fraction (79.99 ± 0.70%), and the chloroform fraction (78.78 ± 2.16%), showing a statistically significant decreasing trend (*p* < 0.05). The ethyl acetate fraction exhibited the highest α-glucosidase inhibitory activity, significantly surpassing the other fractions (*p* < 0.05). This inhibitory effect is likely attributed to moderately polar phenolic and flavonoid compounds, such as polydatin and resveratrol, which can bind to key amino acid residues within the enzyme’s active site. This binding either blocks substrate access or induces conformational changes, resulting in competitive or non-competitive inhibitory activity [31]. Molecular docking studies further corroborate these interactions, demonstrating strong binding affinities between these compounds and the catalytic residues of tyrosinase and elastase, thereby elucidating the potential molecular mechanisms underlying their enzyme inhibitory activities [32]. Regarding lipase inhibitory activity, the *n*-hexane, ethyl acetate, and *n*-butanol fractions exhibited comparable inhibitory activity rates of 85.93%, 85.79%, and 84.35%, respectively (*p* > 0.05). These fractions are believed to contain non-polar terpenoids, anthraquinone derivatives such as emodin-8-O-glucoside, and esterified stilbenoids like polydatin, which likely interact with the enzyme’s catalytic site, interfering with substrate hydrolysis [33]. Overall, the varying polarity of chemical constituents in *P. cuspidatum* suggests multiple mechanisms of enzyme inhibitory activity, highlighting its potential as a natural multi-target metabolic modulator.

### 2.5. Antibacterial Activity of Solvent Fractions from P. cuspidatum Broth Against Selected Microbial Strains

Plant-derived antibacterial agents are mainly categorized as phenolic acids, polyphenols, quinones, flavonoids, flavonols, tannins, terpenoids, and lectins [34]. Although the precise antibacterial mechanisms of these natural products have not been fully elucidated, studies have shown that phenolics and flavonoids inhibit microbial growth by binding essential nutrients such as water and inorganic salts [35]. Additionally, quinones, particularly anthraquinone derivatives, exert antibacterial effects by generating reactive oxygen species (ROS) through redox cycling, which leads to damage to microbial DNA, proteins, and cell membranes [36]. Microorganisms are generally classified into bacteria and fungi. Bacteria are classified as Gram-positive or Gram-negative based on differences in their cell wall structure. Gram-positive bacteria possess a thicker peptidoglycan layer, whereas Gram-negative bacteria have a thinner one. These structural differences result in varying susceptibilities to antibacterial agents [37,38]. Such structural variations may contribute to the differential antibacterial effects observed for *P. cuspidatum* extracts against various microbial strains. The antibacterial activity of solvent fractions from *P. cuspidatum* is presented in Table 5 and Table 6. At a concentration of 5 mg/disc, the ethyl acetate, *n*-hexane, and aqueous fractions exhibited antibacterial activity against *Staphylococcus aureus*. At 10 mg/disc, the ethyl acetate fraction showed antibacterial activity against *Staphylococcus aureus*, *Escherichia coli*, and *Pseudomonas aeruginosa*. In addition, the ethyl acetate, aqueous, *n*-hexane, and *n*-butanol fractions consistently exhibited antibacterial activity against *Staphylococcus aureus*. HPLC-PDA analysis in this study revealed that although the chloroform fraction contained the highest concentration of emodin and relatively low levels of tannic acid, the ethyl acetate fraction exhibited the strongest antibacterial activity. This observation suggests that antibacterial efficacy is not solely dependent on the concentration of individual compounds but may also be influenced by potential synergistic interactions among multiple bioactive constituents. The hydrophobic components in the chloroform fraction may inhibit the release and diffusion of emodin, whereas the matrix of the ethyl acetate fraction appears to facilitate the dissolution and dispersion of various active compounds, including polyphenols and flavonoids, thereby enhancing the overall antibacterial effect. Previous studies have shown that polyphenol compounds such as resveratrol and tannic acid exert antibacterial effects by disrupting bacterial membranes, inhibiting key enzymatic activities, or inducing oxidative stress responses [39]. Moreover, ethyl acetate, a medium-polarity solvent, has polarity characteristics that facilitate the extraction and stabilization of amphiphilic bioactive components. This increases their solubility in aqueous environments and enhances their permeability across cell membranes, thereby improving their bioavailability and antibacterial potency. In summary, these findings emphasize the importance of the polarity distribution of chemical constituents in plant extracts, as well as their compatibility with the polarity of the extraction solvent, in determining antibacterial efficacy. This provides valuable theoretical support for the functional development of natural products.

### 2.6. High-Performance Liquid Chromatography with Photodiode Array Detection (HPLC-PDA) Quantification of Tannic Acid and Emodin

Quantitative analysis of tannic acid and emodin in the solvent fractions of *P. cuspidatum* was performed using HPLC-PDA. The retention times of tannic acid and emodin were 1.71 min and 8.01 min, respectively. Both analytes exhibited baseline separation with symmetrical peak shapes and no tailing, indicating excellent chromatographic resolution and robust method performance. Representative chromatograms and quantification results are shown in Figure 1 and Figure 2.

#### 2.6.1. Evaluation of Linearity, Sensitivity, and Precision

Systematic validation of the developed HPLC-PDA method was performed for linearity, sensitivity, and precision. As summarized in Table 7, tannic acid and emodin exhibited excellent linearity over the concentration range of 25 to 500 ppm, with correlation coefficients (R^2^) of 0.9999 and 0.9995, respectively. The limits of detection (LOD) were 9.24 ppm for tannic acid and 1.82 ppm for emodin, while the limits of quantification (LOQs) were 27.94 ppm and 5.81 ppm, respectively. The relative standard deviations (RSDs) for replicate measurements were below 0.01%, indicating excellent repeatability. Chromatographic analysis showed retention times of 1.72 min for tannic acid (detected at 270 nm) and 7.96 min for emodin (detected at 254 nm), both achieving baseline separation with symmetrical peak shapes and no significant tailing. These findings confirm that the method can reliably quantify the target compounds in *P. cuspidatum* solvent fractions accurately.

#### 2.6.2. Evaluation of Method Precision and Accuracy

Systematic validation of precision and accuracy was conducted for the HPLC-PDA analytical method. As shown in Table 8, tannic acid exhibited excellent repeatability, with intra-day and inter-day relative standard deviations (RSDs) not exceeding 0.03% and 0.01%, respectively. Emodin also demonstrated strong repeatability across all tested concentrations, with intra-day RSD ≤ 0.38% and inter-day RSD ≤ 0.11%. In both cases, inter-day precision was slightly superior to intra-day values, indicating stable method performance over time. Regarding accuracy, tannic acid showed recovery values ranging from 81.78% to 91.82% at 25 ppm, which increased to a range of 94.59% to 103.5% at 50 and 75 ppm. Emodin maintained consistent accuracy across all concentration levels, with recoveries ranging from 99.19% to 109.8%. These results confirm that the developed HPLC-PDA method possesses satisfactory reproducibility and reliability and is suitable for the quantitative determination of target compounds in *P. cuspidatum* solvent fractions.

#### 2.6.3. Quantitative Analysis of Tannic Acid and Emodin by HPLC-PDA

In this study, tannic acid and emodin were quantified in different solvent fractions of *P. cuspidatum* using HPLC-PDA; the results are summarized in Table 9. Tannic acid was found at the highest concentration in the ethyl acetate fraction (0.272 ± 0.004 mg/g), followed by the chloroform fraction (0.105 ± 0.005 mg/g). Lower concentrations were observed in the *n*-hexane (0.028 ± 0.002 mg/g) and *n*-butanol (0.044 ± 0.002 mg/g) fractions, while no tannic acid was detected in the aqueous fraction. Emodin was most abundant in the chloroform fraction (0.270 ± 0.019 mg/g), with moderate levels in the *n*-hexane fraction (0.103 ± 0.003 mg/g). Only trace amounts were found in the ethyl acetate (0.012 ± 0.002 mg/g) and *n*-butanol (0.011 ± 0.003 mg/g) fractions, and it was not detected in the aqueous fraction. The extraction efficiency of tannic acid and emodin varied significantly across different solvent fractions, primarily due to the interplay between solvent polarity and the structural characteristics of each compound. Tannic acid (Figure 3), a hydrolyzable tannin rich in phenolic hydroxyl groups and aromatic rings, exhibits amphiphilic properties. Ethyl acetate, a medium-polarity solvent, yielded the highest extraction (0.272 ± 0.004 mg/g), likely due to its ability to interact with both hydrophilic and hydrophobic regions through hydrogen bonding and π–π interactions [40]. In contrast, the non-polar solvent *n*-hexane exhibited limited solubility, whereas overly polar solvents such as *n*-butanol and water may excessively stabilize hydrophilic moieties, thus impairing overall solubility. Emodin (Figure 4), a hydrophobic anthraquinone with a rigid tricyclic aromatic structure and weak polarity, showed the highest extraction efficiency in the chloroform fraction (0.270 ± 0.019 mg/g). Chloroform’s moderate polarity effectively solubilized the aromatic skeleton of emodin, whereas *n*-hexane, despite its low polarity, tended to extract lipids rather than rigid aromatic compounds. Ethyl acetate and *n*-butanol, being relatively more polar, were not suitable for efficiently dissolving emodin [41]. Both compounds were undetectable in the aqueous fraction, indicating extremely low water solubility. In summary, the optimal extraction of phenolic and anthraquinone-type compounds depends on the precise alignment of solvent polarity with the functional group composition of the target molecules. This understanding provides a theoretical basis for developing efficient extraction protocols for antioxidant, pharmaceutical, and cosmeceutical applications.

### 2.7. Comprehensive Gas Chromatography–Mass Spectrometry Detection GC–MSD Analysis of Solvent Fractions from P. cuspidatum

To systematically characterize the chemical constituents of *P. cuspidatum*, sequential solvent partitioning was performed, followed by analysis using GC–MSD. Five solvent fractions of differing polarity *n*-hexane, chloroform, ethyl acetate, *n*-butanol, and aqueous were analyzed to reveal the polarity-dependent distribution of secondary metabolites. The *n*-hexane fraction exhibited the highest chemical diversity and was predominantly composed of hydrophobic, low-polarity compounds. Notable constituents included silicon-based compounds such as methyltris(trimethylsiloxy)silane (11.94%) and arsenous acid tris(trimethylsilyl) ester (9.57%), both of which have been associated with antibacterial and antidiabetic activities. In addition, minor amounts of cyclic trisiloxanes (3.91%) and flavonoid derivatives such as 1-benzopyrylium, 3,7-dihydroxy-2-(4-hydroxyphenyl) (2.52%), contributed antioxidant and antibacterial properties (Table 10, Figure 5). The chloroform fraction was enriched in potent antioxidant and antibacterial agents, notably 7,9-di-tert-butyl-1-oxaspiro [4,5] deca-6,9-diene-2,8-dione (14.17%), phenol, 2,4-bis(1,1-dimethylethyl) (3.72%), and histamine dihydrochloride (5.54%). These compounds exhibited synergistic antioxidant, anti-inflammatory, and antibacterial activities (Table 10, Figure 5). In the ethyl acetate fraction, histamine dihydrochloride was the predominant constituent (20.76%), suggesting strong antioxidant potential. Additionally, the detection of AICAR (1H-imidazole-4-carboxamide, 5-amino; 3.49%) and 1-benzopyrylium derivatives (7.79%) suggests potential anti-inflammatory and antibacterial properties (Table 10, Figure 1). The *n*-butanol fraction was characterized by elevated levels of 2,5-di-tert-butyl-1,4-benzoquinone (19.10%) and dibutyl phthalate (19.49%), both compounds recognized for their antioxidant and antibacterial activities. Moderate concentration of AICAR (4.30%) and histamine dihydrochloride (7.81%) further contributed the strong bioactivity profile of this fraction (Table 10, Figure 5). The aqueous fraction, as the most polar extract, primarily contained hydrophilic antioxidant and antibacterial constituents, such as 1-benzopyrylium, 3,7-dihydroxy-2-(4-hydroxyphenyl) (17.68%), histamine dihydrochloride (13.96%), and 1,10-phenanthrolinium chloride (13.70%) (Table 10, Figure 5). Overall, the *n*-hexane fraction exhibited the highest diversity of volatile compounds, attributed to its ability to solubilize lipophilic and low-polarity constituents, including waxes, siloxanes, and other lipid-soluble metabolites. In contrast, more polar solvents such as ethyl acetate, *n*-butanol, and water primarily extracted non-volatile or high-boiling-point polar compounds, leading to reduced chemical diversity in GC–MSD analysis. This polarity-driven partitioning behavior underscores the pivotal role of solvent polarity in determining phytochemical extraction efficiency. To comprehensively evaluate both quantitative and qualitative attributes, this study integrated HPLC-PDA quantification of key phenolic and flavonoid markers with in vitro antioxidant, anti-inflammatory, and enzyme inhibitory activity. GC–MSD analysis was subsequently conducted on the most bioactive fractions, enabling a robust correlation between chemical composition and biological activity. This multi-platform analytical strategy provides a scientifically robust foundation for the potential development of *P. cuspidatum* in pharmaceutical, nutraceutical, and cosmeceutical applications.

## 3. Discussion

The ethyl acetate fraction of *P. cuspidatum* exhibited significant antioxidant, anti-diabetic, anti-obesity, skin-whitening, anti-aging, and antibacterial activities in multiple in vitro assays. These biological effects are primarily attributed to the synergistic actions of abundant natural bioactive compounds, particularly phenolics and flavonoids. Qualitative and quantitative analyses using HPLC–PDA and GC–MSD identified key compounds, including tannic acid, emodin, and several benzopyran derivatives, thereby providing strong evidence for the molecular basis of its bioactivities. At the molecular level, the identified compounds exhibited potent free radical scavenging activity and the ability to inhibit reactive oxygen species through electron donation and metal ion chelation, thereby protecting against oxidative stress and preserving cellular integrity. The moderate polarity of the ethyl acetate fraction facilitated the co-extraction of both lipophilic and hydrophilic compounds, enabling effective interactions with multiple target enzymes such as α-glucosidase, lipase, tyrosinase, and elastase. Such interactions, likely mediated through hydrogen bonding and hydrophobic forces, modulate enzymatic activity and contribute to the observed hypoglycemic, anti-obesity, depigmenting, and anti-wrinkle effects. Moreover, the antibacterial effects may result from multiple mechanisms, including disruption of microbial membranes, inhibitory activity of metabolic enzymes, and interference with DNA synthesis. These multifaceted pharmacological activities underscore the potential of the ethyl acetate fraction as a valuable natural source for the development of multifunctional therapeutic agents. Collectively, the findings support the hypothesis that the ethyl acetate fraction exerts its effects via a multi-target, multi-pathway synergistic mechanism. However, this study did not assess the toxicity or safety profile of the ethyl acetate fraction or its isolated compounds. Comprehensive in vitro and in vivo toxicity evaluations are essential to ensure safety before any clinical or commercial application. Future research should prioritize these assessments to fully validate the therapeutic potential of *P. cuspidatum* extracts. Additionally, further investigations employing molecular docking, cellular assays, and in vivo models are warranted to elucidate the precise targets and mechanisms of action of the bioactive constituents.

## 4. Materials and Methods

### 4.1. Chemicals and Plant Material

The *P. cuspidatum* material used in this study was purchased from a local herbal market in Yeongcheon, Gyeongsangbuk-do, Republic of Korea, in September 2024. The dried roots were ground to a particle size of 25–40 mesh using a mechanical grinder and stored in a desiccator until further use.

Gallic acid (Sigma-Aldrich, St. Louis, MO, USA), 1,1-diphenyl-2-picrylhydrazyl (DPPH) (Sigma-Aldrich, St. Louis, MO, USA), 2,4,6-tris(2-pyridyl)-1,3,5-triazine (TPTZ) (Sigma-Aldrich, St. Louis, MO, USA), ABTS diammonium salt (Sigma-Aldrich, St. Louis, MO, USA), L-3,4-dihydroxyphenylalanine (L-DOPA) (Sigma-Aldrich, St. Louis, MO, USA), mushroom tyrosinase (Sigma-Aldrich, St. Louis, MO, USA), N-succinyl-(Ala)_3_-p-nitroanilide (Sigma-Aldrich, St. Louis, MO, USA), porcine pancreatic lipase (Sigma-Aldrich, St. Louis, MO, USA), and 3-(N-morpholino)propanesulfonic acid (MOPS) (Sigma-Aldrich, St. Louis, MO, USA) were all purchased from Sigma-Aldrich (St. Louis, MO, USA). Ferric chloride hexahydrate (FeCl_3_·6H_2_O) (Samchun Pure Chemical Co., Ltd. Seoul, Republic of Korea) and iron(II) sulfate hexahydrate (FeSO_4_·6H_2_O) (Samchun Pure Chemical Co., Ltd. Seoul, Republic of Korea) were obtained from Samchun Pure Chemical Co., Ltd. (Seoul, Republic of Korea). All solvents used for extraction, high-performance liquid chromatography-photodiode array (HPLC-PDA)(YOUNG IN Chromass, Anyang-si, Gyeonggi-do, Republic of Korea), and analytical procedures were of guaranteed reagent (GR) grade and were purchased from Duksan Pure Chemicals Co., Ltd. (Ansan-si, Gyeonggi-do, Republic of Korea).

### 4.2. Extraction and Solvent Fractionation

A total of 200 g of *P. cuspidatum* powder was extracted three times at room temperature with 70% ethanol (*v*/*v*) at a solvent-to-material ratio of 10:1 (*v*/*w*), with each extraction lasting 24 h. After each round, the mixture was filtered through Whatman No. 2 filter paper (Cytiva Ltd., Maidstone, UK). The combined filtrates were subsequently concentrated by freeze-drying using a lyophilizer (Model HFD-1, Huachen Instrument Co., Ltd., Zhengzhou, China) to remove ethanol and obtain the crude ethanolic extract. The dried extract was reconstituted in 200 mL of distilled water and sequentially extracted with various solvents, as illustrated in Figure 6. First, the aqueous solution was mixed with *n*-hexane in a 1:1 (*v*/*v*) ratio and fractionated using a separatory funnel. The organic phase was collected and concentrated under reduced pressure using a rotary evaporator (Model WEV-1001 L, Daihan Scientific Co., Ltd., Seoul, Republic of Korea) at 40 °C. The extraction yielded 3.03% of the *n*-hexane fraction. The remaining aqueous layer was sequentially partitioned with chloroform, ethyl acetate, and *n*-butanol using the same procedure. The extraction yielded 9.61%, 24.48%, and 8.59% of the chloroform, ethyl acetate, and-butanol fractions, respectively. The final aqueous fraction had a yield of 32.78%, as detailed in Table 11. All fractions were stored at temperatures below 4 °C and were dissolved in 70% methanol prior to subsequent bioactivity and chemical analyses.

### 4.3. Total Polyphenol and Flavonoid Contents of Solvent Fractions from P. cuspidatum

The total polyphenol content was measured using the Folin–Denis method [42]. A 200 μL aliquot of the sample was mixed with 200 μL of Folin–Ciocalteu’s phenol reagent (Sigma-Aldrich, St. Louis, MO, USA) and allowed to react for 3 min at room temperature. Then, 3 mL of 10% sodium carbonate was added, and the mixture was incubated in the dark for 60 min. The absorbance was measured at 765 nm using a UV–Vis spectrophotometer (Optizen Pop-s, KLab, Daejeon, Republic of Korea). The total polyphenol content was expressed as mg gallic acid equivalents (GAE) per gram of sample, calculated using a standard curve obtained with gallic acid (Sigma-Aldrich, St. Louis, MO, USA). The total flavonoid content was measured following the method of Jia et al. [43]. A 250 μL aliquot of the sample, diluted with distilled water, was mixed with 1 mL of distilled water and 75 μL of 5% sodium nitrite (*w*/*v*, Samchun Pure Chemical Co., Ltd., Seoul, Republic of Korea). After 5 min, 150 μL of 10% aluminum chloride (*w*/*v*, Junsei Chemical Co., Ltd.) was added, and the mixture was allowed to stand for 6 min. Then, 500 μL of 1 M sodium hydroxide (Junsei Chemical Co., Ltd., Tokyo, Japan) was added, and, after 11 min, the absorbance was measured at 510 nm. The total flavonoid content was expressed as mg quercetin equivalents (QEs) per gram of sample, calculated using a standard curve obtained with quercetin (Sigma-Aldrich, St. Louis, MO, USA).

### 4.4. Antioxidant Activities (DPPH, ABTS Radical Scavenging Activity and Ferric Reducing Antioxidant Power (FRAP) Assay) of Solvent Fractions from P. cuspidatum

The DPPH (1,1-diphenyl-2-picrylhydrazyl) radical scavenging activity was evaluated according to the methods of Blois [44] and others [45]. Briefly, 500 μL of 0.2 mM DPPH solution was mixed with 500 μL of the sample and allowed to react in the dark for 30 min at room temperature. The absorbance was measured at 517 nm using a UV–Vis spectrophotometer (Optizen Pop-s, KLab Inc., Seoul, Republic of Korea). The reduction in absorbance caused by DPPH radical scavenging was assessed. Blank samples were prepared by substituting the sample with the extraction solvent. The DPPH scavenging activity (%) was calculated using the formula below, and the 50% inhibitory concentration (IC_50_) was determined. L-ascorbic acid (Sigma-Aldrich, St. Louis, MO, USA) was used as a reference control for comparison and analysis. The DPPH radical scavenging activity was calculated using the following formula:DPPH radical scavenging activity (%) = (1 − (A/B)) × 100
A stands for “The absorbance of the sample”, and B stands for “The absorbance of the control”.

The ABTS (2,2′-azino-bis-3-ethylbenzothiazoline-6-sulfonic acid) radical scavenging activity was measured following the methods of Pellegrini et al. [46] and others [47]. To prepare the ABTS radical solution, 88 μL of 140 mM potassium persulfate solution was added to 5 mL of distilled water, followed by the addition of two tablets of ABTS diammonium salt (Sigma-Aldrich, St. Louis, MO, USA) to achieve a final concentration of 7 mM ABTS. This solution was left in the dark for 14–16 h, and the resulting ABTS solution was diluted with absolute ethanol at a ratio of 1:88 (*v*/*v*) to adjust the absorbance to 0.70 ± 0.02 at 734 nm. Blank samples were prepared by substituting the sample with the extraction solvent. The ABTS scavenging activity (%) was calculated using the formula below, and the 50% inhibitory concentration (ABTS IC_50_) was determined. L-ascorbic acid was used as positive control for comparison and analysis. The ABTS radical scavenging activity was calculated using the following formula:ABTS radical scavenging activity (%) = (1 − (A/B)) × 100
A stands for “The absorbance of the sample”, and B stands for “The absorbance of the control”.

The ferric reducing antioxidant power (FRAP) assay was measured according to the method of Benzie et al. [48]. The FRAP reagent was prepared by mixing pre-prepared acetate buffer (0.3 M, pH 3.6; Sigma-Aldrich, St. Louis, MO, USA) with 10 mM 2,4,6-tripyridyl-S-triazine (TPTZ; Sigma-Aldrich, St. Louis, MO, USA) and 20 mM ferric chloride solution (Samchun Pure Chemical Co., Ltd., Seoul, Republic of Korea) at a ratio of 10:1:1 (*v*/*v*/*v*). The sample (30 μL), diluted in 70% methanol, was mixed with 900 μL of the FRAP reagent and 90 μL of distilled water. After vortexing, the mixture was left to stand at 37 °C for 10 min, and the absorbance was measured at 593 nm. The FRAP value was expressed as mM of iron sulfate hexahydrate per gram of sample, calculated using a standard curve obtained with iron sulfate hexahydrate (Samchun Pure Chemical Co., Ltd., Seoul, Republic of Korea).

### 4.5. Tyrosinase and Elastase Inhibitory Activities of Solvent Fractions from P. cuspidatum

Tyrosinase inhibitory activity was measured according to the method outlined by Choi et al. [49]. The reaction mixture contained 500 μL of 0.1 M potassium phosphate buffer (pH 6.8), 200 μL of sample, and 200 μL of 10 mM L-DOPA (dihydroxy-phenylalanine, Sigma-Aldrich, St. Louis, MO, USA). Then, 100 μL of enzyme solution (mushroom tyrosinase, Sigma-Aldrich Co.) was added. The mixture was incubated at 37 °C for 15 min, and the change in absorbance at 475 nm was measured. The inhibitory activity was calculated based on the change in dopachrome absorbance. Kojic acid (Sigma-Aldrich, St. Louis, MO, USA) was used as the control. The tyrosinase inhibitory activity was calculated using the following formula:Tyrosinase inhibitory activity (%) = (1 − (A − B)/C) × 100
A stands for “The absorbance at 475 nm determined with sample”, B stands for “The absorbance at 475 nm determined with buffer instead of enzyme”, and C stands for “The absorbance at 475 nm determined with buffer instead of sample”.

Elastase inhibitory activity was measured following the method described by Choi et al. [49]. A mixture of 0.2 M Tris-HCl buffer (pH 8.0) 350 μL and *n*-succinyl-(Ala)_3_-*p*-nitroanilide (Sigma-Aldrich, St. Louis, MO, USA) 125 μL was prepared. Subsequently, 20 μL of the sample was added and 5 μL of elastase (pancreatic from porcine pancreas, PPE, 1.4 units/mg, Sigma-Aldrich Co.) at a concentration of 50 μL/mL was added. The reaction was carried out at 37 °C in an incubator for 20 min. The absorbance was then measured at 410 nm using a UV–Vis spectrophotometer (Optizen Pop-s, KLab Inc., Seoul, Republic of Korea). Elastase inhibitory activity was calculated using the formula below, and ascorbic acid, known for its benefits in aging skin, was used as a control for comparison and analysis. The elastase inhibitory activity was calculated using the following formula:Elastase inhibitory activity (%) = (1 − (A − B)/C) × 100
A stands for “absorbance at 410 nm determined with sample”, B stands for “absorbance at 410 nm determined with buffer instead of enzyme”, and C stands for “absorbance at 410 nm determined with buffer instead of sample”.

### 4.6. α-Glucosidase and Lipase Inhibitory Activities of Solvent Fractions from P. cuspidatum

The α-glucosidase inhibitory activity was evaluated using a modified spectrophotometric method based on the enzyme–substrate reaction, as described by Eom et al. [50]. Briefly, 450 μL of α-glucosidase solution (0.1 U/mL) was mixed with 50 μL of either the sample solution or 0.1 M sodium phosphate buffer (pH 6.8). The mixture was incubated at 37 °C for 15 min. After incubation, 500 μL of 1 mM p-NPG (p-nitrophenyl-α-D-glucopyranoside, Sigma-Aldrich, St. Louis, MO, USA) was added as the substrate and further incubated for 5 min. The amount of p-nitrophenol released by enzymatic hydrolysis was measured at 405 nm using a UV–Vis spectrophotometer (Optizen Pop-s, KLab Inc., Seoul, Republic of Korea). The α-glucosidase inhibitory activity was calculated using the formula below. Acarbose, a well-known α-glucosidase inhibitor, was used as a positive control for comparative analysis. The α-glucosidase inhibitory activity was calculated using the following formula:α-Glucosidase inhibitory activity (%) = (1 − (A/B)) × 100
A stands for “The absorbance of the sample”, and B stands for “The absorbance of the control”.

The lipase inhibitory activity was determined using the method described by Kim et al. [51] with slight modifications. For the preparation of the enzyme buffer, 0.3 mg of porcine pancreatic lipase (triacylglycerol acyl-hydrolase, EC 3.1.1.3; Sigma-Aldrich, St. Louis, MO, USA) was dissolved in 30 μL of 10 mM MOPS (3-[N-morpholino] propanesulfonic acid; Sigma-Aldrich, St. Louis, MO, USA) and 30 μL of 1 mM EDTA (pH 6.8), followed by the addition of 850 μL of Tris buffer (100 mM Tris-HCl, 5 mM CaCl_2_, pH 7.0). After preparation, 100 μL of the sample solution was added to the enzyme buffer and incubated at 37 °C for 15 min. Subsequently, 20 μL of 10 mM p-nitrophenyl palmitate (p-NPP; Sigma-Aldrich, St. Louis, MO, USA) was added to initiate the enzymatic reaction, which was further incubated at 37 °C for an additional 15 min. The degree of hydrolysis of p-NPP to p-nitrophenol was quantified by measuring absorbance at 400 nm using a UV–Vis spectrophotometer (Optizen Pop-s, KLab Inc., Seoul, Republic of Korea). The lipase inhibitory activity was calculated using the formula below. Orlistat, a clinically approved lipase inhibitor, was used as a positive control for comparative analysis. The lipase inhibitory activity was calculated using the following formula:Lipase inhibitory activity (%) = (1 − (B − b)/A) × 100(1)
A stands for “absorbance at 400 nm determined with buffer instead of sample”, B stands for “absorbance at 400 nm determined with sample”, and b stands for “absorbance at 400 nm determined with buffer instead of enzyme”.

### 4.7. Antibacterial Activities of Solvent Fractions from P. cuspidatum Broth Against Selected Microbial Strains

The antibacterial activities of solvent fractions derived from *P. cuspidatum* broth was evaluated using the disc diffusion assay, as previously described by Blois [52]. Seven microbial strains were selected for the antibacterial activity testing, including Gram-positive bacteria: *Staphylococcus aureus* KCTC 1621 and Gram-negative bacteria: *Escherichia coli* KCTC 1112 and *Pseudomonas aeruginosa* KCTC 2450. These strains were obtained from the Korean Collection for type culture (KCTC). The bacterial strains used for antibacterial activity were cultured under the conditions specified in Table 12. Each strain was inoculated into nutrient broth and cultured at 30 °C and 37 °C for 24–30 h through three subcultures. The absorbance at 600 nm was measured, and strains with absorbance values falling within the range of 0.2 to 0.4 (corresponding to 1 × 10^5^ CFU/mL) were selected for antibacterial activity testing. For antibacterial activity testing, the activated strains were evenly spread on nutrient agar to prepare the test plates. Test solutions were prepared for each sample to achieve concentrations of 5 and 10 mg per disc. Paper discs (8 mm) were impregnated with each sample solution, dried to evaporate the solvent, and then placed in close contact with the agar surface of the test plates. After incubating at 30 °C or 37 °C for 24–30 h, the diameter (mm) of the clear zone around each disc, indicating inhibitory activity against microbial growth, was measured to compare antibacterial efficacy.

### 4.8. HPLC-PDA Quantification of Tannic Acid, and Emodin

Quantitative analysis of gallic acid, tannic acid, and emodin was performed using a YOUNG IN Chromass ChroZen HPLC system (YOUNG IN Chromass, Anyang-si, Gyeonggi-do, Republic of Korea) equipped with a quaternary pump, autosampler, and photodiode array (PDA) detector. Separation was performed using a Poroshell 120 EC-C18 column (4.6 × 150 mm, 4 μm; Agilent Technologies, Santa Clara, CA, USA). For tannic acid, isocratic elution was performed using a mobile phase consisting of 50% solvent A (0.5% (*v*/*v*) acetic acid in water) and 50% solvent B (methanol) over 0–20 min. Detection was conducted at 270 nm, with a flow rate of 1.0 mL/min, column temperature set at 30 °C, and injection volume of 10 μL. For emodin, isocratic elution was carried out using 75% solvent A (0.1% (*v*/*v*) phosphoric acid in water) and 25% solvent B (methanol) over 30 min. Detection was performed at 254 nm, with the same flow rate, column temperature, and injection volume as described above.

### 4.9. HPLC-PDA Method Validation

The analytical method was validated in accordance with the guidelines of European Commission Decision 2002/657/EC. The validation parameters included linear range, calibration curves, sensitivity, precision, and accuracy.

#### 4.9.1. Linearity, Sensitivity, and Precision

Linearity was assessed by analyzing a series of standard working solutions at various concentrations using the established HPLC-PDA method. Calibration curves were generated by plotting peak area (Y) against analyte concentration (X), with the correlation coefficient (R^2^) indicating reliable quantification within the validated range. Sensitivity was evaluated by determining the limits of detection (LODs) and quantification (LOQs), based on signal-to-noise ratios of 3 and 10, respectively. Low-concentration spiked samples were analyzed to estimate these limits. The LOQ values met the acceptance criteria for recovery (>70%) and precision (RSD ≤ 20%), confirming the method’s reliability at low concentrations. Precision was evaluated by calculating the relative standard deviation (RSD) for both intra-day and inter-day repeatability. Intra-day precision was assessed by analyzing three replicate samples at different concentrations within a single day, while inter-day precision was determined by repeating the same procedure over three consecutive days. The method was considered acceptably precise if the RSD was less than 15%.

#### 4.9.2. Accuracy and Specificity

Accuracy was determined through recovery experiments by spiking known amounts of analytes into blank matrices at three concentration levels (25–500 ppm). Each level was analyzed in six replicates following the same sample preparation procedure. Recovery rates (%) were calculated by comparing the measured concentrations to the theoretical spiked values, ensuring the method’s trueness and reliability. Specificity was confirmed by analyzing blank matrices, standard solutions, and matrix-spiked samples. Chromatographic results demonstrated that the target analytes were well resolved under the optimized conditions, with no interference from endogenous matrix components, indicating a high selectivity.

### 4.10. GC–MSD Quantification of Solvent Fractions from P. cuspidatum

GC–MSD analysis was conducted using an Agilent 5975 GC–MSD system equipped with an HP-5ms capillary column (30 m × 0.25 mm i.d., 0.25 μm film thickness; Agilent Technologies, Santa Clara, CA, USA). Helium was used as the carrier gas at a constant flow rate of 1.0 mL/min. The GC oven temperature was initially held at 60 °C for 3 min, then ramped to 240 °C at 5 °C/min, and further increased to 300 °C, where it was held for 5 min. The injector temperature was set to 230 °C. Mass spectra were acquired in electron ionization (EI) mode at 70 eV.

### 4.11. Statistical Analysis

All experiments were conducted with measurements repeated at least three times, and the results were expressed as mean ± standard error. Statistical analysis was performed using the SPSS Statistics 22.0 software system (ANOVA, SPSS Inc., Chicago, IL, USA). Analysis of variance (ANOVA) was carried out to validate the results. For significant findings, Duncan’s multiple range test was employed for post hoc analysis at a significance level of *p* < 0.05.

## 5. Conclusions

In conclusion, the ethyl acetate fraction of *P. cuspidatum* demonstrated the most diverse and potent bioactivities among all solvent fractions, including antioxidant, anti-inflammatory, enzyme inhibitory, and antibacterial effects. These bioactivities are closely associated with the presence of phenolic and flavonoid compounds identified through HPLC–PDA and GC–MSD analyses. These findings provide a scientific basis for developing this fraction as a potential active ingredient in functional foods, pharmaceuticals, and cosmetic formulations. Further studies are warranted to elucidate the in vivo efficacy and precise molecular targets of these bioactive constituents.

## Figures and Tables

**Figure 1 ijms-26-07011-f001:**
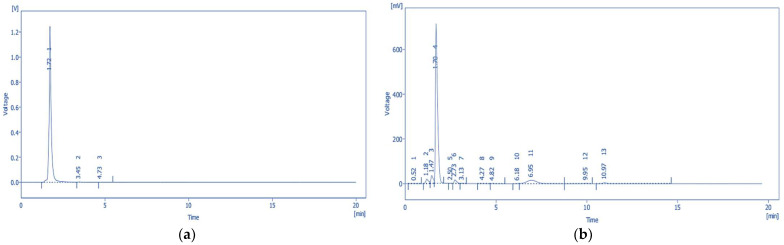
Chromatographic and photodiode array (PDA) spectral data of tannic acid: (**a**) HPLC chromatogram of the tannic acid standard solution showing 3 peaks; (**b**) chromatogram of the *P. cuspidatum* solvent fractions showing 13 peaks; (**c**) PDA spectrum of the tannic acid standard; and (**d**) PDA spectrum of the corresponding sample peak. The identical spectral profiles confirm the presence of tannic acid in the sample matrix.

**Figure 2 ijms-26-07011-f002:**
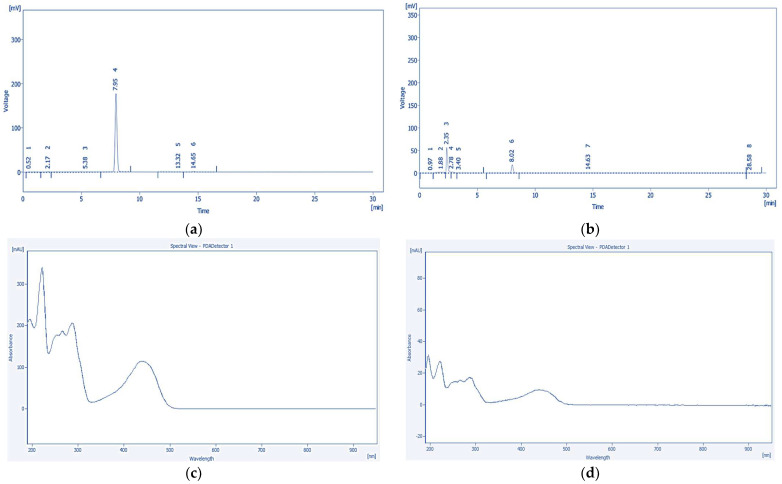
Chromatographic and photodiode array (PDA) spectral data of emodin: (**a**) HPLC chromatogram of the emodin standard solution showing 6 peaks; (**b**) chromatogram of the *P. cuspidatum* solvent fractions showing 8 peaks; (**c**) PDA spectrum of the emodin standard; and (**d**) PDA spectrum of the corresponding sample peak. The identical spectral profiles confirm the presence of emodin in the sample matrix.

**Figure 3 ijms-26-07011-f003:**
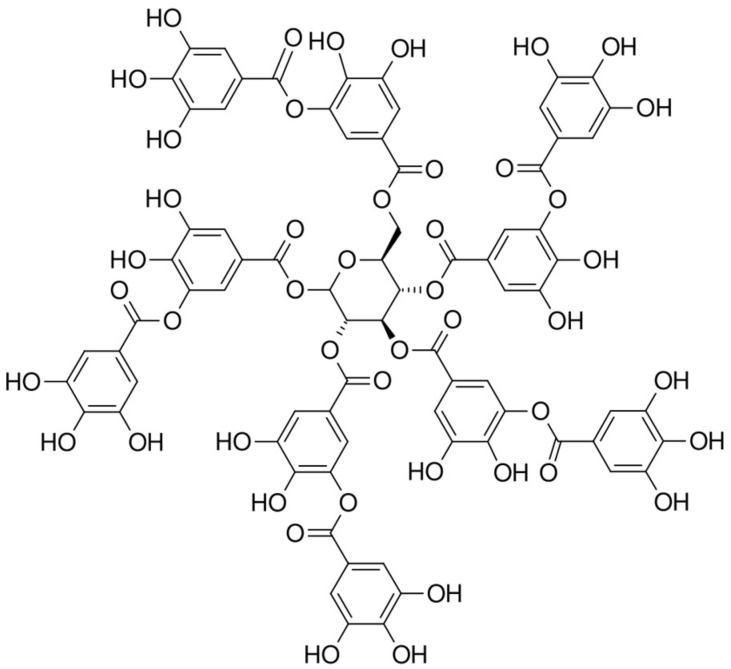
Chemical structure of tannic acid, a representative hydrolyzable tannin characterized by multiple phenolic hydroxyl groups and a polyaromatic framework.

**Figure 4 ijms-26-07011-f004:**
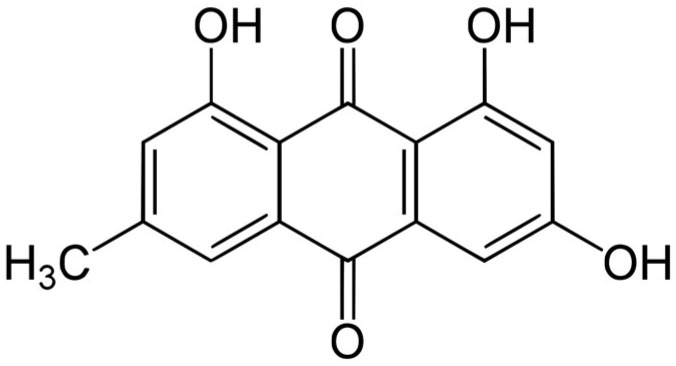
Chemical structure of emodin, a hydrophobic anthraquinone with a rigid tricyclic aromatic structure and weak polarity.

**Figure 5 ijms-26-07011-f005:**
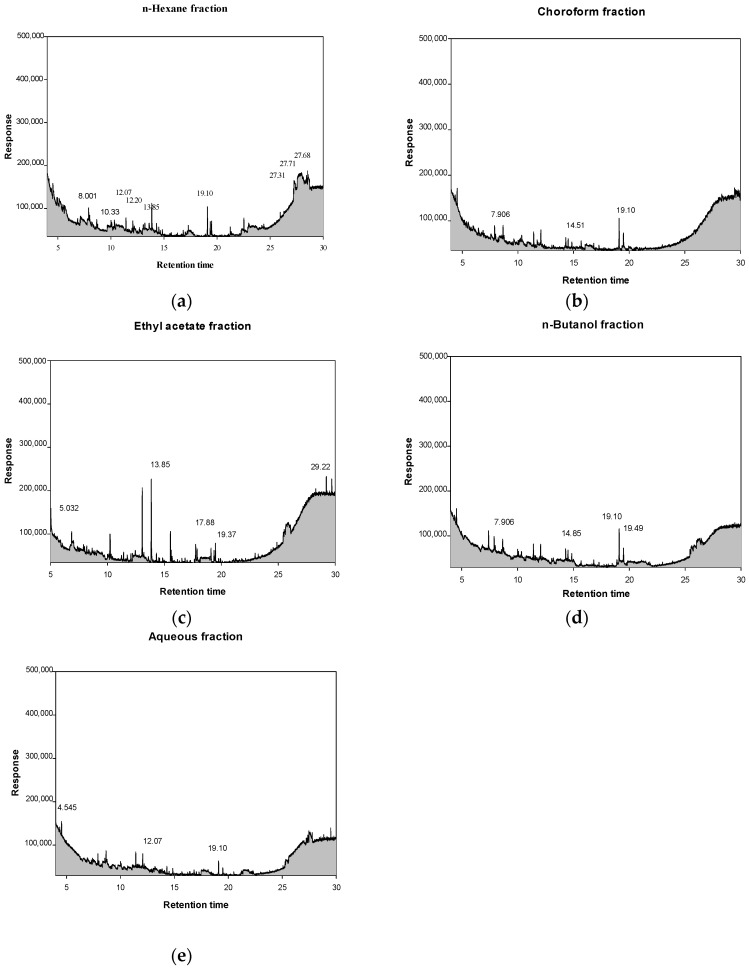
GC–MSD chromatograms profiles of *P. cuspidatum* solvent fractions (**a**) *n*-hexane fraction, (**b**) chloroform fraction, (**c**) ethyl acetate fraction, (**d**) *n*-butanol fraction, and (**e**) aqueous fraction.

**Figure 6 ijms-26-07011-f006:**
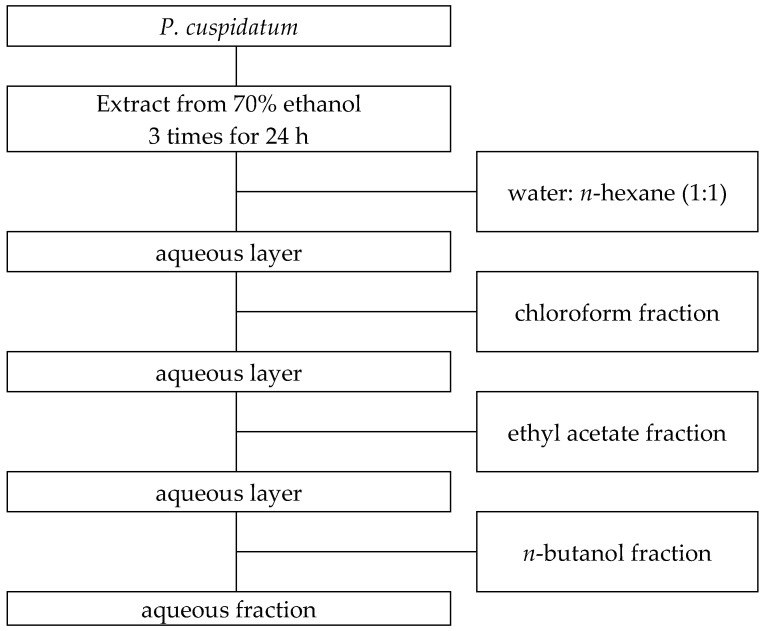
Stepwise solvent fractionation scheme of the 70% ethanolic extract of *P. cuspidatum*. The dried extract was successively partitioned with *n*-hexane, chloroform, ethyl acetate, and *n*-butanol, resulting in the corresponding solvent fractions.

**Table 1 ijms-26-07011-t001:** Total polyphenol and flavonoid contents of solvent fractions from *P. cuspidatum*.

Solvent	Total Polyphenol Content (g GAE/g) ^1^	Total Flavonoid Content (g QE/g) ^2^
*n*-hexane fraction	0.12 ± 0.00 ^c34^	0.08 ± 0.00 ^c34^
chloroform fraction	0.37 ± 0.01 ^b^	0.09 ± 0.00 ^c^
ethyl acetate fraction	0.53 ± 0.01 ^a^	0.19 ± 0.02 ^a^
*n*-butanol fraction	0.36 ± 0.02 ^b^	0.14 ± 0.01 ^b^
aqueous fraction	0.10 ± 0.01 ^d^	0.04 ± 0.00 ^d^

^1^ GAE: gallic acid equivalent mg/g. ^2^ QE: quercetin equivalent mg/g. ^3^ Mean ± SD (n = 3). ^4 a–d^ Means Duncan’s different letters within a column differ significantly (*p* < 0.05).

**Table 2 ijms-26-07011-t002:** DPPH and ABTS radical scavenging activities and ferric reducing antioxidant power (FRAP) values of solvent fractions from *P. cuspidatum*.

Solvent	DPPH Radical Scavenging Activity IC_50_ (mg/mL) ^1^	ABTS Radical Scavenging Activity IC_50_ (mg/mL) ^1^	FRAP Value (mM Fe^2+^/mg)
*n*-hexane fraction	0.25 ± 0.20 ^a23^	0.64 ± 0.19 ^a23^	0.22 ± 0.05 ^d23^
chloroform fraction	0.07 ± 0.00 ^c^	0.14 ± 0.00 ^c^	1.84 ± 0.14 ^c^
ethyl acetate fraction	0.01 ± 0.00 ^d^	0.06 ± 0.00 ^d^	6.02 ± 0.30 ^a^
*n*-butanol fraction	0.02 ± 0.00 ^d^	0.13 ± 0.00 ^c^	2.95 ± 0.57 ^b^
aqueous fraction	0.10 ± 0.01 ^b^	0.51 ± 0.00 ^b^	0.32 ± 0.02 ^d^
L-ascorbic acid	0.01 ± 0.00 ^d^	0.06 ± 0.00 ^d^	-

^1^ Inhibitory activity was expressed as the mean of the 50% inhibitory concentration from triplicate determinations, obtained by interpolation of the concentration-inhibition curve. ^2^ Mean ± SD (n = 3). ^3 a–d^ Means Duncan’s different letters within a column differ significantly (*p* < 0.05).

**Table 3 ijms-26-07011-t003:** Tyrosinase and elastase inhibitory activities of solvent fractions from *P. cuspidatum*.

Solvent	Tyrosinase Inhibitory Activity (%)	Elastase Inhibitory Activity (%)
*n*-hexane fraction	59.93 ± 1.95 ^c14^	68.69 ± 1.31 ^bc14^
chloroform fraction	53.60 ± 1.61 ^d^	64.65 ± 1.50 ^c^
ethyl acetate fraction	67.78 ± 2.50 ^a^	83.84 ± 1.64 ^a^
*n*-butanol fraction	61.64 ± 0.65 ^b^	74.75 ± 1.74 ^b^
aqueous fraction	56.31 ± 0.86 ^cd^	72.73 ± 0.60 ^bc^
standard	69.26 ± 0.10 ^e2^	63.64 ± 0.59 ^c3^

^1^ Mean ± SD (n = 3). ^2^ Kojic acid. ^3^ L-Ascorbic acid. ^4 a–e^ Means Duncan’s different letters within a column differ significantly (*p* < 0.05).

**Table 4 ijms-26-07011-t004:** α-Glucosidase and lipase inhibitory activities of solvent fractions from *P. cuspidatum*.

Solvent	α-Glucosidase Inhibitory Activity (%)	Lipase Inhibitory Activity (%)
*n*-hexane fraction	5.560 ± 0.59 ^d14^	85.93 ± 1.00 ^b14^
chloroform fraction	12.87 ± 0.15 ^c^	78.78 ± 2.16 ^c^
ethyl acetate fraction	65.14 ± 0.29 ^a^	85.79 ± 0.61 ^b^
*n*-butanol fraction	35.87 ± 0.56 ^b^	84.35 ± 0.77 ^b^
aqueous fraction	6.460 ± 0.40 ^d^	79.99 ± 0.70 ^c^
standard	34.83 ± 0.31 ^b2^	98.18 ± 0.18 ^a3^

^1^ Mean ± SD (n = 3). ^2^ Acarbose. ^3^ Orlistat. ^4 a–d^ Means Duncan’s different letters within a column differ significantly (*p* < 0.05).

**Table 5 ijms-26-07011-t005:** Antibacterial activity date of solvent fractions from *P. cuspidatum*.

Microorganism	Solvent	5 mg/disc	10 mg/disc
*Staphylococcus aureus*	*n*-hexane fraction	10.0 ^1^	12.0
	chloroform fraction	- ^2^	-
	ethyl acetate fraction	17.0	19.5
	*n*-butanol fraction	-	10.0
	aqueous fraction	9.50	12.5
*Escherichia coli*	*n*-hexane fraction	-	-
	chloroform fraction	-	-
	ethyl acetate fraction	-	10.0
	*n*-butanol fraction	-	-
	aqueous fraction	-	-
*Pseudomonas aeruginosa*	*n*-hexane fraction	-	-
	chloroform fraction	-	-
	ethyl acetate fraction	-	10.0
	*n*-butanol fraction	-	-
	aqueous fraction	-	-

^1^ Size of clear zone (mm). ^2^ Not detected.

**Table 6 ijms-26-07011-t006:** Graphical representation of the antibacterial activities of solvent fractions from *P. cuspidatum*.

Microorganism	*n*-Hexane Fraction	Chloroform Fraction	Ethyl Acetate Fraction	*n*-Butanol Fraction	Aqueous Fraction
*Staphylococcus aureus*	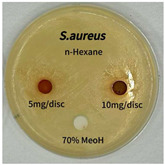	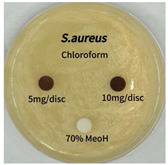	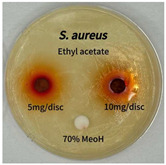	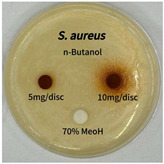	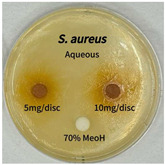
*Escherichia coli*	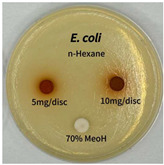	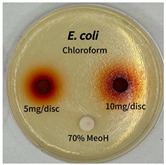	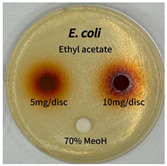	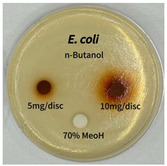	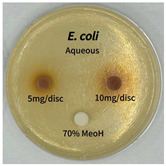
*Pseudomonas aeruginosa*	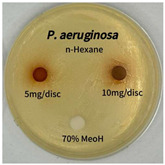	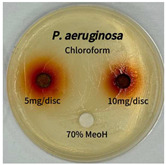	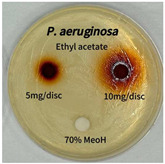	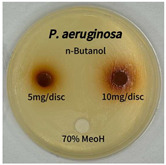	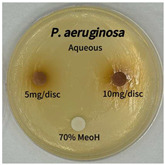

**Table 7 ijms-26-07011-t007:** Retention times, linear range, calibration curves, limits of detection (LODs), and limits of quantitation (LOQs) of tannic acid and emodin compounds.

Parameter	Tannic Acid	Emodin
λ ^1^ (nm)	270	254
t_R_ ^2^ (min)	1.72	7.96
RSD ^3^ (%)	0.01	0.01
Linear range (ppm)	25–500	25–500
Regression equation ^4^	Y = 30.43845X − 550.88152	Y = 3.94432X + 33.84235
r ^2 5^	0.9999	0.9995
LOD (ppm)	9.24	1.82
LOQ (ppm)	27.94	5.81

^1^ Detection wavelength. ^2^ Retention times. ^3^ Relative standard deviation. ^4^ Y: peak area, X: concentration (ppm). ^5^ Coefficient of determination.

**Table 8 ijms-26-07011-t008:** Precision and accuracy of the HPLC-PDA method for tannic acid and emodin compounds.

Compound	Conc. ^1^ (ppm)	Analysis type	Measured Conc. ^2^ (ppm)	RSD ^3^ (%)	Accuracy ^4^ (%)	RSD (%)
Tannic acid	25	Intra-day (n = 3)	27.53	0.03	81.78	0.01
	50		45.45	0.02	97.60	0.03
	75		73.80	0.03	103.5	0.04
	25	Inter-day (n = 3)	29.14	0.01	91.82	0.02
	50		47.96	0.01	94.59	0.01
	75		72.30	0.00	102.2	0.00
Emodin	25	Intra-day (n = 3)	23.23	0.38	99.19	0.57
	50		49.80	0.29	106.3	0.23
	75		78.14	0.16	101.0	0.18
	25	Inter-day (n = 3)	24.17	0.05	102.0	0.09
	50		50.49	0.05	109.8	0.06
	75		79.88	0.09	107.4	0.11

^1^ Theoretical concentration of the compound. ^2^ Measured concentration of the compound. ^3^ Relative standard deviation. ^4^ Percentage recovery of the compound.

**Table 9 ijms-26-07011-t009:** HPLC-PDA analysis of tannic acid and emodin compounds in solvent fractions from *P. cuspidatum*.

Solvent	Tannic Acid (mg/g)	Emodin (mg/g)
*n*-hexane fraction	0.028 ± 0.002 ^d12^	0.028 ± 0.002 ^d^
chloroform fraction	0.105 ± 0.005 ^b^	0.105 ± 0.005 ^b^
ethyl acetate fraction	0.272 ± 0.004 ^a^	0.272 ± 0.004 ^a^
*n*-butanol fraction	0.044 ± 0.002 ^c^	0.044 ± 0.002 ^c^
aqueous fraction	- ^3^	-

^1^ Mean ± SD (n = 3). ^2 a–d^ Means Duncan’s different letters within a column differ significantly (*p* < 0.05). ^3^ Not detected.

**Table 10 ijms-26-07011-t010:** Chemical constituents identified in the solvent fractions derived from *P. cuspidatum* by GC–MSD analysis.

Compound	t_R_ ^1^ (min)	Chemical Class	Major Compounds (Relative Abundance, %)	Reported Biological Activity
*n*-hexane fraction	8.001	Flavonoids	1-Benzopyrylium, 3,7-dihydroxy-2-(4-hydroxyphenyl) (2.52%)	Antibacterial
	10.33	Biogenic amines	Histamine dihydrochloride (1.39%)	Antioxidant
	12.07	Nucleotide intermediates	1H-Imidazole-4-carboxamide, 5-amino(AICAR) (1.60%)	Anti-inflammatory, Antioxidant
	12.20	Nitrogen heterocycles	1,10-phenanthrolinium chloride (1.45%)	Antioxidant, Antibacterial
	13.85	Phthalate esters	1,2-benzene-dicarboxylic acid, dimethyl ester (3.71%)	Antioxidant
	19.10	Spirocyclic diketones	7,9-Di-tert-butyl-1-oxaspiro (4,5) deca-6,9-diene-2,8-dione (3.40%)	Antioxidant
	27.31	Organoarsenic compounds	Arsenous acid, tris(trimethylsilyl) ester (9.57%)	Antidiabetic
	27.68	Organosilicon compounds	methyltris(trimethylsiloxy)silane (11.94%)	Antibacterial
	27.71	Cyclic siloxanes	Cyclotrisiloxane, hexamethy (3.91%)	Antioxidant, Antibacterial
chloroform fraction	7.906	Biogenic amines	Histamine dihydrochloride (5.54%)	Antioxidant
	14.51	Phenolic antioxidants	2,4-Di-tert-butylphenol (3.72%)	Antibacterial, Anti-inflammatory
	19.10	Spirocyclic diketones	7,9-Di-tert-butyl-1-oxaspiro (4,5) deca-6,9-diene-2,8-dione (14.17%)	Antioxidant
ethyl acetate fraction	5.032	Flavonoids	1-Benzopyrylium, 3,7-dihydroxy-2-(4-hydroxyphenyl) (7.79%)	Antibacterial
	13.85	Biogenic amines	Histamine dihydrochloride (20.76%)	Antioxidant
	17.88	Nucleotide intermediates	1H-Imidazole-4-carboxamide, 5-amino(AICAR) (3.49%)	Anti-inflammatory, Antioxidant
	19.37	Nitrogen heterocycles	1,10-phenanthrolinium chloride (3.61%)	Antioxidant, Antibacterial
	29.22	Phthalate esters	1,2-benzene-dicarboxylic acid, dimethyl ester (2.99%)	Antioxidant
*n*-butanol fraction	7.906	Biogenic amines	Histamine dihydrochloride (7.81%)	Antioxidant
	14.85	Nucleotide intermediates	1H-Imidazole-4-carboxamide, 5-amino(AICAR) (4.30%)	Anti-inflammatory, Antioxidant
	19.10	Quinones	2,5-di-tert-butyl-1,4-benzoquinone (DTBBQ) (19.10%)	Antioxidant, Antibacterial
	19.49	Phthalate esters	Dibutyl phthalate (19.49%)	Antibacterial
aqueous fraction	4.545	Nitrogen heterocycles	1,10-phenanthrolinium chloride (13.70%)	Antioxidant, Antibacterial
	12.07	Biogenic amines	Histamine dihydrochloride (13.96%)	Antioxidant
	19.10	Flavonoids	1-Benzopyrylium, 3,7-dihydroxy-2-(4-hydroxyphenyl) (17.68%)	Antibacterial

^1^ Retention times.

**Table 11 ijms-26-07011-t011:** Extraction yields of solvent fractions from *P. cuspidatum*.

Solvent	Yield (%, *w*/*w* ^1^)
*n*-hexane fraction	3.03
chloroform fraction	9.61
ethyl acetate fraction	24.5
*n*-butanol fraction	8.59
aqueous fraction	32.8

^1^ yield (%) was calculated against the dried *P. cuspidatum*.

**Table 12 ijms-26-07011-t012:** List of strains used for antibacterial experiments.

Microorganism	Media ^1^	Temperature (°C)	
*Staphylococcus aureus*	NA/NB	30	
*Escherichia coli*	NA/NB	30	
*Pseudomonas aeruginosa*	NA/NB	37	

^1^ NA: nutrient agar, NB: nutrient broth.

## Data Availability

All data generated during this study are included in this article.

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
