# Peer review of "Solvent Fractionation of Polygonum cuspidatum Sieb. et Zucc. for Antioxidant, Biological Activity, and Chromatographic Characterization"

_ijms, 2025, doi:10.3390/ijms26147011_

Round 1
Reviewer 1 Report
Comments and Suggestions for Authors
Dear authors,
I have just read the manuscript authored by Cheng et al. with interest and made some key comments which need to be resolved.
Abstract
Please add some of the obtained results in this section.
Results
Table 1: it is of high interest that hexane fraction exhibited higher TPC and TFC values than aqueous phase. It seems that medium-polarity solvents are preferable, but most polyphenols are rather waster-soluble than oil-soluble, as well. Kindly compare results with other similar studies to support this evidence.
Please correct to "α-Glucosidase" in subsection 2.4.
Please re-estate sentences 106-108. DPPH is a synthetic radical which is not present in living organisms.
Discussion
Slighlty reinforce this section.
Materials and Methods
Some chemicals are missing from the corresponding subsection.
Italicize (v/v), "p" in N-succinyl-(Ala)₃-p-nitroanilide and "n" in n-hexane and n-succinyl-, and provide uniformity in the manuscript.
Change "minutes" to "min" and "hours" to "h".
References
The cited references are relevant, but some of them are outdated. Kindly replace them with more recent. Limitations of this study could be inserted in a section.
Author Response
Author's Reply to the Review Report (Reviewer 1)
Dear authors,
I have just read the manuscript authored by Cheng et al. with interest and made some key comments which need to be resolved.
Abstract
Please add some of the obtained results in this section.
Answer for abstract : Thank you for the suggestion. I have updated the abstract with the obtained results as requested(Page 1, line 18~26).
Results
Table 1: it is of high interest that hexane fraction exhibited higher TPC and TFC values than aqueous phase. It seems that medium-polarity solvents are preferable, but most polyphenols are rather waster-soluble than oil-soluble, as well. Kindly compare results with other similar studies to support this evidence.
Answer for Table 1 results : Thank you for your comments. We have made the requested revisions accordingly(Page 3, line 105~111, Page 24 Reference 48 addition).
Please correct to "α-Glucosidase" in subsection 2.4.
Answer for subsection 2.4 results : Thank you for your comments. We have made the requested revisions accordingly(Page 5, line 200~201, 236.237, 599, 610).
Please re-estate sentences 106-108. DPPH is a synthetic radical which is not present in living organisms.
Answer for results : Thank you for your comments. We have made the requested revisions accordingly(Page 3, line 119~121, 142-158, Page 22, Reference 18 change).
Discussion
Slighlty reinforce this section.
Answer for discussion : Thank you for your suggestion. We have slightly reinforced the Discussion section as requested(Page 15-16, line 450~460).
Materials and Methods
Some chemicals are missing from the corresponding subsection.
Italicize (v/v), "p" in N-succinyl-(Ala)₃-p-nitroanilide and "n" in n-hexane and n-succinyl-, and provide uniformity in the manuscript.
Change "minutes" to "min" and "hours" to "h".
Answer for materials and methods : The requested changes have been made throughout the entire manuscript(Page 16, line 468, 477, 484, 493, 499, 519-522, 530, 531, 540, 555, 556,568,587,610,658,661).
References
The cited references are relevant, but some of them are outdated. Kindly replace them with more recent. Limitations of this study could be inserted in a section.
Answer for references : Thank you for your valuable suggestions. We have added a section acknowledging the limitations of the toxicity and safety evaluation of the ethyl acetate fraction and emphasizing the need for future toxicological studies. We have also added recent references while retaining the existing references, as they provide essential basic context for the experimental methods and theoretical concepts used in this study(Page 16, line 454-460, Page 24, Reference 48-53 addition ).

Reviewer 2 Report
Comments and Suggestions for Authors
International Journal of Molecular Sciences
Manuscript number ijms-3754561
The manuscript entitled “Antioxidant, Biological Activity and Chromatographic Characterization of Polygonum cuspidatum Sieb. et Zucc Natural Products by Solvent Fractionation” evaluated the bioactive potential of P. cuspidatum as well as different interesting inhibitory activities with great importance in human health. This is a well-written and well-structured work that provides a novelty in relation to P. cuspidatum.
Some questions:
- The titled should be modify
- I recommend separating in two paragraphs (for tyrosine inhibition and for elastase inhibition) section 2.3. for a better understanding of.
- Write properly in italics the names of bacteria in the whole section 2.4.
- A more extended discussion of the results should be needed. It is an important lack of references and comparison with other literature works.
- Please, correct the word “inhibitory” in all section 4.5
Author Response
Author's Reply to the Review Report (Reviewer 2)
The manuscript entitled “Antioxidant, Biological Activity and Chromatographic Characterization of Polygonum cuspidatum Sieb. et Zucc Natural Products by Solvent Fractionation” evaluated the bioactive potential of P. cuspidatum as well as different interesting inhibitory activities with great importance in human health. This is a well-written and well-structured work that provides a novelty in relation to P. cuspidatum.
Some questions:
- The titled should be modify
Answer : Thank you for the suggestion. I corrected the title.(Page 1, line 2~4).
- I recommend separating in two paragraphs (for tyrosine inhibition and for elastase inhibition) section 2.3. for a better understanding of.
Answer : Thank you for the suggestion. I have separated Section 2.3 into two paragraphs on tyrosine inhibition and elastase inhibition for better understanding.(Page 4-5, 166-194 ).
- Write properly in italics the names of bacteria in the whole section 2.4.
Answer : Thank you for the suggestion. I have edited all bacteria names into italics.(Page 7, line 258~260).
- A more extended discussion of the results should be needed. It is an important lack of references and comparison with other literature works.
Answer : Thank you for the suggestion. The results were discussed in more detail, and a reference was added.(Page 15-16, line 450~460, Page 24, line 820-834).
- Please, correct the word “inhibitory” in all section 4.5
Answer : Thank you for the suggestion. I changed all of 4.5-4.6 to the word "inhibitory"(Page 18-19, 574,575, 580, 582, 586, 592, 594, 595, 599, 600, 610, 610, 613, 625, 626).
